# Automated Lane Centering: An Off-the-Shelf Computer Vision Product vs. Infrastructure-Based Chip-Enabled Raised Pavement Markers

**DOI:** 10.3390/s24072327

**Published:** 2024-04-05

**Authors:** Parth Kadav, Sachin Sharma, Johan Fanas Rojas, Pritesh Patil, Chieh (Ross) Wang, Ali Riza Ekti, Richard T. Meyer, Zachary D. Asher

**Affiliations:** 1Department of Mechanical and Aerospace Engineering, Western Michigan University, 4601 Campus Dr, Kalamazoo, MI 49008, USA; parth.kadav@wmich.edu (P.K.); sachin.sharma@wmich.edu (S.S.); priteshyashaswi.patil@wmich.edu (P.P.); richard.meyer@wmich.edu (R.T.M.); 2Revision Autonomy Inc., 4717 Campus Drive, Kalamazoo, MI 49008, USA; johan.fanasrojas@wmich.edu; 3Oak Ridge National Laboratory, Oak Ridge, TN 37831, USAektia@ornl.gov (A.R.E.)

**Keywords:** autonomous vehicles (AV), advanced driver assistance systems (ADAS), chip enabled raised pavement marker (CERPM), perception, controls, vehicle-to-infrastructure (V2I), infrastructure sensors, lane centering, Mobileye, computer vision

## Abstract

Safe autonomous vehicle (AV) operations depend on an accurate perception of the driving environment, which necessitates the use of a variety of sensors. Computational algorithms must then process all of this sensor data, which typically results in a high on-vehicle computational load. For example, existing lane markings are designed for human drivers, can fade over time, and can be contradictory in construction zones, which require specialized sensing and computational processing in an AV. But, this standard process can be avoided if the lane information is simply transmitted directly to the AV. High definition maps and road side units (RSUs) can be used for direct data transmission to the AV, but can be prohibitively expensive to establish and maintain. Additionally, to ensure robust and safe AV operations, more redundancy is beneficial. A cost-effective and passive solution is essential to address this need effectively. In this research, we propose a new infrastructure information source (IIS), chip-enabled raised pavement markers (CERPMs), which provide environmental data to the AV while also decreasing the AV compute load and the associated increase in vehicle energy use. CERPMs are installed in place of traditional ubiquitous raised pavement markers along road lane lines to transmit geospatial information along with the speed limit using long range wide area network (LoRaWAN) protocol directly to nearby vehicles. This information is then compared to the Mobileye commercial off-the-shelf traditional system that uses computer vision processing of lane markings. Our perception subsystem processes the raw data from both CEPRMs and Mobileye to generate a viable path required for a lane centering (LC) application. To evaluate the detection performance of both systems, we consider three test routes with varying conditions. Our results show that the Mobileye system failed to detect lane markings when the road curvature exceeded ±0.016 m^−1^. For the steep curvature test scenario, it could only detect lane markings on both sides of the road for just 6.7% of the given test route. On the other hand, the CERPMs transmit the programmed geospatial information to the perception subsystem on the vehicle to generate a reference trajectory required for vehicle control. The CERPMs successfully generated the reference trajectory for vehicle control in all test scenarios. Moreover, the CERPMs can be detected up to 340 m from the vehicle’s position. Our overall conclusion is that CERPM technology is viable and that it has the potential to address the operational robustness and energy efficiency concerns plaguing the current generation of AVs.

## 1. Introduction

Autonomous vehicle (AV) technology has the potential to improve safety by using advanced sensors to map the vehicle’s surroundings. Additionally, data processing helps the AV make rapid decisions, aiding drivers in complex traffic, detecting obstacles, minimizing risks by reducing human errors, and enabling new energy-efficient operational strategies. However, research efforts are still required to fully optimize their potential. AVs traditionally use inputs from an array of on-vehicle sensors, such as cameras, radio detection and ranging (RADAR), global navigation satellite system (GNSS), and light detection and ranging (LiDAR) [1]. These sensors are used for environmental perception, such as vehicle detection and tracking, pedestrian detection, road surface detection, road lane detection, and traffic sign detection. The data from these sensors are processed using advanced fusion algorithms where the outputs are used in the planning and control subsystems. The information gathered from sensors is used by the current state-of-the-art fusion and machine-learning algorithms. This typically requires an onboard computer with high operating speeds and/or multiple processors [2]. Advanced driver assistance systems (ADAS) are the current widespread commercially available application of AV technology. By leveraging the on-vehicle sensors, ADAS provides specific functionality, such as lane keeping assist (LKA), lane centering (LC), automated emergency braking (AEB), and much more. These systems aid the driver by providing proactive safety and comfort [2,3].

ADAS is set to be mandated by the National Highway Traffic Safety Administration (NHTSA) for all passenger vehicles and light trucks by 2030 [4]. These systems rely on onboard sensors and computations for real-time applications. For example, LC, a Society of Automotive Engineers (SAE) designated level 2 automation, also known as auto-steer, is a system that is responsible for keeping the vehicle in the center of a lane [5]. ADAS, such as LC and LKA, utilizes computer vision (CV) techniques to process the information and use it for vehicle control. For example, Mobileye, which is a common component used in ADAS products for LC and LKA, provides computer vision processing. Mobileye can perform LKA and LC when clear, visible lane markings are present [6]. Typical lane detection technology based on computer vision utilizes image processing algorithms. These algorithms extract features of lane lines by reducing image channels, processing acquired images, and fitting lane lines after extraction [7,8]. Computer vision systems, although useful, have limitations. They may struggle in bad weather or low light, and obstacles can block their view. Poor lane markings and lane line obstructions decrease their reliability and performance as well [9]. Moreover, the sensors on the vehicle only have information about the immediate surroundings within their limited range. There is a notable shortfall in the performance of vehicle automation products when navigating sharp curves, other extreme road geometries, poor environmental conditions, and poor lane markings [10,11,12]. Moreover, in tackling the limitations of computer vision systems, contemporary ADAS incorporates neural network-based detection algorithms. While these algorithms offer enhanced robustness and accuracy, they come with the trade-off of increased computational demands that translate to an additional load on the vehicle’s computer and difficulty overcoming instances of failure [13]. In response to these drawbacks, we have previously undertaken the development of a novel, infrastructure-based technology [14]. This technology aims to transmit environmental information directly to the vehicle using infrastructure-based sensors.

Wireless communications between infrastructure-based sensors enable vehicles to exchange vital information through vehicle-to-vehicle (V2V) and vehicle-to-infrastructure (V2I) applications [15]. The effectiveness of on-vehicle sensors is subject to external factors like road conditions, lighting, lane markings, and weather. For instance, vision-based lane detection systems may encounter difficulties when environmental conditions fluctuate, such as changes in illumination, shadows, or inclement weather [16]. Other research from our lab has shown that onboard sensors and onboard computation can have significant toll on the vehicle’s range and efficiency [17]. This study will thoroughly examine one of these solutions. Infrastructure-based sensors can transmit information directly to the vehicle via wireless communications. This V2I data can enhance the performance of existing perception systems and facilitate the development of novel methods for vehicles to perceive their surroundings [18,19]. Roads currently use raised pavement markers (RPMs) that are placed along lane markings to provide visual guidance for human drivers. These markers are made of reflective materials that help with nighttime driving and are maintained by state Departments of Transportation (DOTs) as a low-cost, passive solution. They also have a snow-plowable alternative that can withstand winter treatments [20]. In our previous work, we developed a new way of gathering information for vehicle subsystems using infrastructure information sources (IISs) [14]. We created chip-enabled raised pavement markers (CERPMs). These are a modified version of standard RPMs, designed to provide crucial information for the perception systems in AVs through V2I communications. CERPMs are V2I sensors that provide static geospatial information to surrounding vehicles. The CERPMs are small, cost-effective, and low-power IISs that have a long transmission range and are suitable for on-road use. The CERPMs operate in the United States unlicensed industrial, scientific, and medical (ISM) radio frequency band of 915 MHz. They can enable cooperative driving automation (CDA). In a subsequent study, we tested a new strategy for AV lane-keeping using inputs from a camera sensor and the CERPMs [14]. CERPMs were set up in the CARLA simulator to simulate real-world sensor placement scenarios. We estimated the lateral offset using the data simulated in the CARLA simulator, and all tests were performed in the simulator itself. Previous work has also been performed to investigate the performance and operational impacts of CERPMs [14].

Although the simulation tests were useful, they worked under ideal conditions and did not show the entire picture. Additionally, no controls were applied to the vehicle in the study, and further testing would be required to test the LC/LKA using the CERPM technology. To address the shortcomings of previous research, conducting a proof-of-concept vehicle integration by testing the CERPMs with a vehicle in real-world conditions, and incorporating an AV subsystem equivalent, is needed. In this novel study, we deployed the CERPMs on actual roads and designed various test scenarios to evaluate their viability as V2I sensors for real-world vehicle applications. The perception subsystem utilizes the CERPMs and minimal data processing to generate the necessary trajectory for the test route, followed by control for LC. We compared the performance of the CERPMs with the Mobileye 6 Development Kit for LC, a commercially available off-the-shelf computer vision solution [21]. The primary contributions are as follows:Implementing the initial proof-of-concept CERPMs on actual roads;Real-time data processing and vehicle integration for LC;Comparing V2I against traditional on-vehicle lane line detection methods for LC;On-road testing for vehicle control using CERPMs for LC.

## 2. Methodology

Initially, the focus here is on the perception subsystem and recent advancements since the last studies  [14,22]. Having gathered information from the perception systems using either the Mobileye or the CERPM, the next step is to proceed to the control subsystem and implement vehicle control. Both the CERPMs and Mobileye will be assessed for their detection performance on two different test routes with varying curvatures. The detections, which act as inputs to the perception subsystem, will then be used to create a reference trajectory for vehicle control. This trajectory will be evaluated for controller error. The tests were all conducted in clear and sunny conditions to obtain unobscured and clear detections for the Mobileye. So overall, the performance in terms of perception and controls will be evaluated in different test cases based on established drive cycles for the two different systems.

### 2.1. AV Subsystem: Perception

The research studies presented in this work were conducted using the energy-efficient autonomous vehicle (EEAV) laboratory research vehicle platform, as shown in Figure 1. This platform is a 2019 Kia Niro that has been equipped with both sensor-based perception and control subsystems. These additions include a forward-facing stereo camera, two global navigation satellite systems (GNSSs) receivers, a Polysync DriveKit that enables drive-by-wire functionality, and the Mobileye.

The Polysync DriveKit connects to the controller area network (CAN) of the vehicle, which can accurately read and control throttle, brake, and steering. Additionally, the drive kit comes with its own robot operating system (ROS) driver, which facilitates communication between the various on-vehicle sensors. A detailed discussion about each sensor and its setup are provided in the following sections.

#### 2.1.1. On-Vehicle Sensor: GNSS

GNSS is a general technology used for vehicle positioning on land. To precisely estimate the position of the ego-vehicle in the world, two GNSS receivers along with real-time kinematics (RTK) were utilized. The instrumented research vehicle is equipped with one SwiftNav Duro Inertial RTK kit and one SwiftNav Duro RTK kit placed on the roof of the vehicle, as shown in Figure 2a, providing us with 2-centimeter accuracy using SwiftNav’s Skylark Precise Positioning service. As shown in Figure 2b, the Duro Inertial RTK is placed in the front, which includes a GNSS receiver and an inertial measurement unit (IMU), and the Duro RTK is placed behind the Duro Inertial RTK. This enables heading measurements and more accurate localization. Swift Navigation also provides the ROS drivers for these sensors. Different topics are published in the ROS environment including the position, orientation, heading, and accuracy of the receivers placed on the vehicle.

#### 2.1.2. On-Vehicle Sensor: Mobileye Camera

The Mobileye system used for this study is the 6 Development Kit, as shown in Figure 3a. The Mobileye was placed on the windshield, as shown in Figure 3b. Mobileye is an advanced driver assistance system that includes a high dynamic range complementary metal-oxide-semiconductor (CMOS HDRC) camera, on-board image processing, and also supports a ROS driver. Following the mounting of the Mobileye, it is calibrated using a thorough calibration process based on its placement, both for images and communication with the vehicle’s CAN bus, allowing it to fetch information from the vehicle.

*Setup* 

The Mobileye is connected to the vehicle CAN bus using a Kvaser USB-can adapter [23]. The adapter offers two high-speed CAN channels and one USB 2.0 interface that enables communication between the Mobileye, on-vehicle computer, and the vehicle CAN bus. Kvaser provides drivers and a software development kit (SDK) for Linux, which is written in both Python and C, and enables configuration and communication protocol. The Mobileye does not provide the raw image data, but provides information such as lane boundaries, dynamic obstacles, traffic signs, collision warnings, lane departure warnings, and headway information. The Mobileye 6 Development Kit is developed for roads that have clear and visible lane markings on both sides of the lane. The Mobileye can operate in low lighting conditions and during adverse weather conditions, but the performance is not as robust. It can accurately detect the fully visible bicycles and rear end of cars as well as fully visible pedestrians.

*Data Routing for AV System* 

The ROS driver developed by Autonomous Stuff integrates the sensor with the ROS ecosystem, offering crucial information such as lane data for detected lanes including confidence value, offset distance, type, and curvature [24]. The lane information and published data can be used for developing the lane model and the reference trajectory. The published information in the ROS workspace can be processed using custom algorithms for the perception subsystem. The output from the perception task can be further used for vehicle control. Figure 4 shows the overall Mobileye data routing plan.

*Data Processing* 

The information published by the Mobileye ROS drivers contains topics that can be used for obtaining information such as detected lane positions, offset distance, and lane curvature, which can be processed and used for the perception subsystem. We used the topic as_tx/lane_markers, which provides visualization and location information about the detected lanes. The detected lane markings with their corresponding identification number, type, and positions are published in a local coordinate frame that is centered and originated at the sensor. If no lanes are detected, then no detections are published to the topic. The detections follow a right-handed coordinate system, where the positive X-axis points straight ahead, The positive Y-axis points to the left and the positive Z-axis points up. The different lane model types outputted by the topic are linear, parabolic, and 3D. Since these calculations are handled internally by Mobileye, we will leverage the information provided by the ROS driver, such as the lane models, and generate a reference trajectory. We can obtain the reference trajectory by obtaining the lane offsets from the left lane line and the right lane line individually and computing the center. Figure 5a shows the output from the Mobileye for a single lane using RViz, which is a visualization tool for ROS. This is for visualization purposes only and does not accurately represent distances in terms of scale or position. Starting from the left, the green line represents the left lane line marking, the second green line is the right lane line marking, and the red line is the lane boundary. The center of the current lane, i.e., the reference trajectory, is shown using the purple line in Figure 5b. The reference trajectory is taken as the path needed for vehicle control and further compared to the CERPM path.

*Preparation for Lane Centering (LC) Using Mobileye* 

The ego-vehicle is to travel within the center of the lane defined by the reference trajectory as mentioned in Section 2.1.2. Using the trajectory information via the perception subsystem, we would apply control by passing the control output to the control subsystem, so that the ego-vehicle follows the center of the lane. We will discuss this in detail in Section 2.3.2.

### 2.2. Chip Enabled Raised Pavement Markers

As established in our previous work, we developed a new method of transmitting lane information to on-road vehicles using sensors that act as active IIS using V2I technology [22]. Low-cost sensors, referred to as CERPMs, were developed. RPMs shown in Figure 6a were utilized with modifications to transmit geospatial information such as latitude, longitude, and altitude. Various tests with CERPMs were conducted both at the Western Michigan University and the Oak Ridge National Laboratory [14,22]. In this section, the CERPM technology is expanded for real-world vehicle perception and control tasks.

*Component Selection and Communication Protocol* 

Using long-range (LoRa), and a low-power wide-area network (LPWAN) protocol, an IoT development board was added to the RPMs to transmit information. The transmitter (Tx) was placed in the given RPM, as shown in Figure 6b. An IoT development board called WiFi LoRa 32 was chosen as it is integrated with the SX1276 LoRa Connect transceiver that supports industrial, scientific, and medical (ISM) band of data transmission frequencies [25]. The chip also allows the support of the Arduino library. A rechargeable 3.7 V Lithium battery of 1000 mAh is connected to power the IoT board. This powers the IoT board. An antenna is attached to the board to help transmit the information. The antenna does not require external power and is powered through the IoT board itself. The Tx, along with the antenna and the battery enclosed in the RPMs, are referred to as CERPMs. The receiver (Rx) was set up similarly, however, it was modified from the original antenna by AEACAQ190012-S915 [26] for increased antenna gain. The Tx LoRa nodes transmit the information in form of radio packets from the LoRa Tx node to the Rx LoRa node. The Rx also receives information such as the receiver signal strength index (RSSI), and signal-to-noise ratio (SNR). Both the in-vehicle Rx and CERPMs were programmed to operate at a radio frequency of 915 MHz, which is an unlicensed ISM radio frequency band.

*CERPM Setup* 

Geospatial data were collected using the Trimble Catalyst DA2, a GNSS positioning sensor, in the World Geodetic System 1984 (WGS 84) coordinate system. This data consist of latitude, longitude, altitude, and additional parameters, such as horizontal accuracy, vertical accuracy, precision, point identification, and timestamp. The sensor has a maximum precision of 2 cm in horizontal accuracy and 2 cm in vertical accuracy. The CERPMs were programmed to transmit the geospatial information. Additionally, each CERPM was assigned a unique identification number for tracking and processing purposes.

*Data Routing for AV System* 

Figure 7 shows the plan for AV control using the CERPMs. A serial communication from the CERPM was established with the on-vehicle computer via the CP2102 USB to serial chip on the IoT board using the Rx.

Data packets received from the Rx were read using the serial library in a Python script. Unlike the Mobileye, a custom ROS driver had to be created to publish the obtained information from the Txs to the ROS environment for the perception subsystem. The Rx, when connected to the in-vehicle computer, sends the received packets using the serial port to the computer. The data within these packets can be extracted using a Python script in conjunction with the serial library. Subsequently, this information can be decoded and transmitted to the ROS environment via the same Python script, thereby enabling accessibility to all other processes within the ROS master. The CERPM ROS driver operates at 20 hz, which is every 0.05 s.

*Data Processing* 

By utilizing the CERPM information collected through the ROS driver, the CERPMs can be processed based on their geospatial information. To establish a feasible vehicle path using GNSS data, the CERPMs are categorized into the following two groups: CERPMs belonging to the left lane boundary and the right lane boundary. This separation is based on whether they are placed on the left lane line or the right lane line within a given lane. However, due to the limited number of CERPMs placed at specific intervals along the pavement, interpolation became crucial to fill the gaps between these markers. Interpolation ensures a complete set of CERPMs on both sides, compensating for any missing CERPMs due to signal dropout. After separating the CERPMs into left and right lane edges and performing cubic interpolation, the reference trajectory for the vehicle is derived. The cubic spline interpolation polynomial $S_{i}\left(x\right)$ for the interval $\left[x_{i},x_{i+1}\right]$ is given by:(1)$$S_{i}\left(x\right)=a_{i}+b_{i}\left(x-x_{i}\right)+c_{i}{\left(x-x_{i}\right)}^{2}+d_{i}{\left(x-x_{i}\right)}^{3}$$
where $a_{i}$, $b_{i}$, $c_{i}$, and $d_{i}$ are coefficients determined by the interpolation and continuity conditions, and *i* is the number of points used for interpolation.

The interpolated CERPM information is transformed into local coordinates about the North-East-Down coordinate frame (NED) relative to the ego vehicle’s position and orientation. This is achieved using the pymap3d library in Python. Transforming geospatial information maps everything local to the vehicle. This transformation generates a local path for the ego-vehicle to follow. The reference trajectory is obtained by calculating the midpoints between the interpolated CERPMs. The resulting reference trajectory represents the local coordinates (x,y,z) of the path that the vehicle should follow.

*Pseudo Lane-Line Projection for Visual Verification* 

To verify the accurate conversion of the geospatial information from the global coordinate frame to the local coordinate frame, as discussed in Section 2.2, the local lane line coordinates are projected on the forward camera feed based on the vehicle’s current position and orientation.

The goal was to transform and align the local coordinates to the camera placed on the vehicle, which could be then overlaid on the camera feed obtained from the ZED2i camera sensor. To perform projection, the local coordinates were shifted from the GNSS base station to the ZED2i using an extrinsic matrix. The extrinsic matrix includes the translation and rotation information between two given sensor frames. Now that all the local coordinates have been translated to the camera frame, the points were converted from the NED coordinate frame to the OpenCV coordinate frame [27].

Next, the points need to be rotated and aligned with the heading (θ) of the ego vehicle. In summary, the coordinate frame was rotated and adjusted with θ along the down axis, which is represented in Equation (Equation 2). Our previous study presents a comprehensive methodology for the material discussed in this section [28].
(2)$$\left[\begin{array}{c} X_{c}\\ Y_{c}\\ Z_{c} \end{array}\right]=\left[\begin{array}{ccc} -\sin\theta & \cos\theta & 0\\ 0 & 0 & 1\\ \cos\theta & \sin\theta & 0 \end{array}\right]\left[\begin{array}{c} N\\ E\\ D \end{array}\right]$$

The points were now in the required coordinate frame, rotated, and aligned. The next step was the projection of these points onto the camera feed. This required determining the pixel location (u, v) for each point, which was performed using the camera intrinsic matrix obtained from the camera data. The 3D points were appropriately projected onto the 2D image plane, using their determined pixel locations. A detailed methodology for this has been established in our previous work [28].

The resulting image has the CERPMs projected as points onto the raw camera feed and can be further used for visual verification. This method does not require the lane lines to be visible or a direct line of sight to the lane. Figure 8a shows the projection of four CERPMs on the camera feed. The left lane markings, represented with blue markers, are the two non-interpolated CERPMs placed at a separation of 40 feet, whereas the two CERPMs on the right lane boundary are interpolated to have more points in between. Figure 8b shows the generation of pseudo lane lines using a total of 10 CERPMs on the right lane boundary and ten CERPMs on the left lane boundary at a separation of 40 feet.

*Preparation for Lane Centering (LC) Using CERPMs* 

In Section 2.2, raw information from the CERPMs was received using the ROS driver. The information was processed using the perception subsystem, which created a reference path in local coordinates. This local path was then used as the final reference trajectory for the AV control subsystem’s lateral control calculation, as discussed in Section 2.3.2.

### 2.3. AV Subsystem: Controls

For LC using either the Mobileye or the CERPM derived reference trajectory, a lateral controller is required, which keeps the vehicle centered in the given lane of travel. For our study, we obtained the lane-line data from either Mobileye or CERPMs. A lateral controller is used to adjust the ego-vehicle’s trajectory based on the offset between its actual position and the reference trajectory. The control output is used to align the vehicle in the given lane. Additionally, a longitudinal controller consider the recommended speed limit for the route, ensuring the ego-vehicle maintains the desired speed, which would be the maximum allowable speed limit for the given route. Figure 9 presents a brief flow diagram demonstrating how perception data from Mobileye or CERPMs can be used to perform LC.

#### 2.3.1. Fixed Longitudinal Controller

The ego-vehicle was set to follow a fixed target speed, which was the speed limit for the test route. The DriveKit published the wheel speed information from all four wheels, using the wheel speed encoder data, which essentially served as the perception subsystem. Using the wheel speed information, the ego-vehicle speed was calculated and published to the ROS environment. For control calculation, a proportional-integral-derivative (PID) controller was designed. The PID controller is a standard control algorithm that is a single-input single-output system (SISO). The PID uses the error between the reference input and the current input to drive it to zero. The system then uses the output from the PID as its control input. The difference between the current speed and the target speed, i.e., the error (Equation (Equation 3)), was the input to the PID, the output from the PID was used to adjust the throttle and brake. This was achieved by sending throttle and brake requests to the vehicle using the DriveKit.
(3)error=targetspeed−currentspeed

If the output is positive that means that the targetspeed is more than currentspeed, in which case we send a throttle request and the vehicle maintains the target velocity. If the output is negative, the vehicle may either coast or brake, depending on the output.

#### 2.3.2. Lateral Controller

The lateral controller is responsible for minimizing the variance between the current position and the reference trajectory. The Stanley controller was implemented for lateral control. It is a path-tracking algorithm designed and implemented by Stanford University’s DARPA Grand Challenge team [29]. The Stanley controller algorithm uses the front axle of the vehicle as the reference point. It minimizes both the heading and the cross-track error for the given trajectory. The cross-track error is the distance between the closest point on the reference trajectory with respect to the vehicle’s front axle. The heading error is the direction the vehicle should be facing on the given trajectory versus the current heading. Figure 10 shows the Stanley controller diagram. ψ denotes the angle between the trajectory heading and the vehicle’s heading, δ represents the steering angle of the vehicle, *v* stands for the velocity of the vehicle, *L* is the wheelbase, and ce is the cross-track error, which is the error between the center of the front axle and the closest point on the path.

Equation (Equation 4) computes the desired steering wheel angle for the ego-vehicle to follow the trajectory.
(4)$$\delta \left(t\right)=\psi \left(t\right)+\arctan\left(\frac{k\times ce(t)}{v(t)}\right)$$
where ψ(t) is the heading error, *k* denotes a smoothing constant, ce(t) stands for the cross-track error, and v(t) represents the ego-vehicle velocity. The standard Stanley controller was modified using a look-ahead distance to the controller, as shown in Equation (Equation 5). The look-ahead distance $d_{l}$ is the distance on the reference trajectory, at which the cross-track error is computed.
(5)$$\delta \left(t\right)=\psi \left(t\right)+\arctan\left(\frac{k\times ce(t,d_{l})}{v(t)}\right)$$

Adding a look-ahead distance gives the control system knowledge about the trajectory at $d_{l}$ meters in front of the vehicle, providing a smoother response. $d_{l}$ was set to be a constant value for both LC systems. Figure 11 shows an overall systems-level diagram for both longitudinal and lateral control outputs.

Similar to the PID controller developed in Section 2.3.1, a low-level PID controller is implemented to minimize the cross-track error and obtain the desired steering wheel angle. The DriveKit has access to the motor-driven power steering (MDPS) system on the EEAV research platform. The output from the PID is then utilized to send steer requests to the MDPS until the error is minimized.

*LC Using Mobileye* 

When clear lane lines are detected on both sides of the lane, the perception subsystem computes the reference trajectory, and in turn, the cross-track error that is used as an input to the lateral control algorithm. The look-ahead distance ($d_{l}$) is set to be a fixed point (15 m) on the reference trajectory. The maximum $d_{l}$ on the reference trajectory is 31 m. Since the Mobileye detections are processed and obtained in a local frame, as mentioned in Section 2.1.2, a fixed look-ahead distance is used to compute the offset. A fixed target speed was set to be achieved by the longitudinal controller, which is the maximum allowable speed limit for the test route.

*LC Using CERPM* 

The local coordinates obtained from Section 2.2 are used as the reference trajectory for LC. In contrast to the Section 2.1.2, where the path moves with the ego-vehicle, the CERPM reference trajectory includes the first measurement from the GNSS on the ego-vehicle as the origin and the rest of the path measured with respect to the origin. To include $d_{l}$, the position of the ego-vehicle is projected 15 m ahead using the vehicle heading. This enables the calculation of cross-track error at $d_{l}$. The computed cross-track error is used as an input to compute the control output and necessary steer requests are sent to the DriveKit to center the ego-vehicle in the given lane. A fixed target speed was set to be achieved by the longitudinal controller, which is the maximum allowable speed limit for the test route.

## 3. Test Routes

The two different sensor technologies, namely, the Mobileye and the CERPMs were used for LC. The tests were conducted at two different locations. Initial testing for the CERPMs was conducted at Oak Ridge National Laboratory (ORNL), which would be the first test route. The second and third tests were conducted on specific sections of a fixed route at Western Michigan University’s (WMU) Parkview Campus Drive shown in Figure 12a. The testing routes were divided into sections based on different road conditions, such as road curvature and cardinal directions. Figure 12b shows the fixed test route used to conduct the different drive cycles, which contains different road curvatures and variance in lane markings. For initial testing, only specific sections of the entire test route are used, streamlining the testing process and maintaining brevity for performance evaluation.

The first test route was at ORNL to investigate the use of CERPM technology for on-road applications and the generation of a reference trajectory for LC. The second test route was a selected section of the WMU campus drive with a steep curvature. Lastly, the third test route was a portion of the WMU campus drive with a low curvature. It has been observed that there is a notable difference in the performance of vehicle automation products on arterial roads with steep road curvatures, uneven road surface finish, improper lane markings, and lane marking obstructions [30]. The CERPM technology can be tested to address these drawbacks. The first test route was established at ORNL to investigate CERPMs for on-road trajectory generation and LC. The second test route at WMU consisted of a steep curvature road section. Lastly, the third test route at WMU consisted of a low curvature road section.

### 3.1. Route 1: Oak Ridge National Laboratory

An initial test was conducted to test on-road trajectory generation using CERPMs. For this purpose, 10 CERPMs were set up on a section of the Old Bethel Valley Road at ORNL. The geospatial data were collected using the Trimble DA2 catalyst and programmed into each CERPM. The selected test route was unpaved and unmarked. The Mobileye was not tested due to the need for lane markings. This preliminary study aimed to generate the reference trajectory required for the control subsystem and evaluate the accuracy of Section 2.2 without first using testing on marked roads.

The CERPMs are placed on the designated test routes following the guidelines provided by the US Department of Transportation’s (USDOT) Federal Highway Administration (FHWA) [31]. There are different regulations for the placement depending on the road type, such as urban, arterial, interstate, on-ramp, and off-ramp. Our tests were mainly conducted on arterial and two-lane roads for initial proof-of-concept testing. The CERPMs were placed on either side of the lane that the ego-vehicle would be traveling in with a specific separation between each CERPM. Depending on the length of the test route, the number of CERPMs and the separation between them was altered for initial testing. The test included the generation of the reference trajectory by placing the ego-vehicle at point A of the route, as shown in Figure 13a. The interpolated reference trajectory generated by the CERPMs was plotted on an open-street map to verify its global position shown in Figure 13b. Additionally, tests were conducted for received signal strength and transmission range distance, as discussed in Section 4.1.

### 3.2. Route 2: WMU Campus Drive Loop Steep Curvature

After initial testing at ORNL, we tested the CERPM for LC alongside the Mobileye at WMU’s test route 2, which was the steep curvature test route that has the starting point A and ending point B, as shown in Figure 14, which had a length of 275 m. The campus drive loop comprises two one-way lanes, with lane markings on both sides of the outermost (right) lane, as shown in Figure 14. For this test, we tested both Mobileye and CERPM LC on the outermost lane of the test route. We utilized the previously collected high-definition lane-line data as the geospatial information necessary to program the CERPMs [28]. In total, 10 CERPMs were placed on either side of the lane across from each other and the raw data were passed onto the perception subsystem for creating the reference path. Similarly, Mobileye data were passed to the perception subsystem for processing and reference path generation. The perception output was sent to the control subsystem for LC using (1) The Mobileye and (2) CERPMs. The performance of both the Mobileye and the CERPMs for steep curvature is discussed in Section 4.2.

### 3.3. Route 3: WMU Campus Drive Low Curvature

Figure 15 shows the low curvature test route. The test was conducted from point A to point B. The length of this section was 350 m. The CERPMs were placed as mentioned in Section 3.2. This analysis provides insights into the adaptability of both the Mobileye and the CERPMs in roads characterized with low curvature. The performance of both the Mobileye and CERPMs for LC in low curvature scenarios is discussed in Section 4.3.

## 4. Results

The results section discusses the performance of the two LC systems for the three different test routes. For test route 1, only CERPMs were tested for initial verification of reference trajectory generation for real-world on-road purposes. Along with the generation of the reference trajectory, the received signal strength indicator (RSSI) was analyzed for test route 1. The reference trajectory trace was compared with the ground truth GNSS lane-line data collected using the Trimble DA2 catalyst. The RSSI, which indicates the signal strength at the Rx side in decibel milliwatts (dBm), will be analyzed first, followed by the results from the Mobileye and the CERPM LC tests conducted at WMU’s Campus Drive.

### 4.1. Signal Strength Analysis

To profile the distance with RSSI for a given CERPM, one CERPM was placed on test route 1 and the RSSI was measured from point A towards point B. The CERPM was measured to be at a distance of 50 m from the starting position, which is point A where the ego-vehicle was present. The RSSI measured −108 dBm at point A and −125 dBm at a distance of 340 m towards point B.

Figure 16 shows a 3D plot of the CERPM’s location relative to the ego-vehicle and the RSSI. The red data marker represents the CERPM’s position in global coordinates. The dynamic Z-axis distance line illustrates the ego-vehicle’s position and its changing proximity to the CERPM, along with the RSSI mapped to the colorbar.

### 4.2. Steep Curvature

The CERPMs and the Mobileye were tested on WMU’s campus drive where visible lane markings were present. This section will look at the steep curvature scenario, which is test scenario 2. CERPMs were set up as mentioned in Section 3.1. After processing the data and generating a reference trajectory as explained in Section 2.2, the processed CERPM data were sent to the control subsystem for both lateral and longitudinal control using only CERPMs.

#### 4.2.1. Mobileye

Figure 17 shows the trajectory generated for the steep curvature test route using Mobileye detections. Along with ground truth data collected in our previous work [28]. Mobileye encountered difficulties in detecting both left and right lane lines for the steep curvature test route. The Mobileye could only successfully detect the right lane line marking for 81.7% of the given route. The detection rate for the left lane line was only 6.7%. The system could detect both the left and right lane line markings for just 6.7% throughout the entire route. It was seen that the Mobileye failed after curvatures greater than 0.016 m^−1^. LC could not be tested due to inadequate detections, preventing the successful generation of a reference trajectory.

#### 4.2.2. CEPRM

The raw CERPM data for both the left and right lane lines of the road is depicted in Figure 18a. Figure 18b shows the generated reference trajectory using CEPRMs and ground truth information during the steep curvature test. After interpolation, we could successfully generate a reference trajectory for the entire path. The generated path was passed onto the control subsystem for lateral control. As compared to the Mobileye, the CERPMs could successfully generate the entire path for the given test route.

#### 4.2.3. LC Performance

Due to insufficient detections from Mobileye, a reference trajectory could not be generated; thus, LC using Mobileye could not be tested, as shown in Figure 19a. LC cannot be engaged at the starting position of the test. Figure 19b shows the mean squared error (MSE) for the path taken by the controller using the CERPM reference trajectory compared to the ground truth. The MSE in X-direction was 0.25 m and the MSE in Y-direction was 0.34 m.

### 4.3. Low Curvature

The CERPMs and the Mobileye were tested on test route 3, which was the low curvature scenario. This section will look at the detections obtained from the two systems.

#### 4.3.1. Mobileye

Figure 20 shows the generated reference trajectory for the low curvature test route along with the ground truth. The Mobileye could successfully detect the right lane line marking (solid white) for 93% of the route, and the left lane line (dashed white) for 86.1%. Both the left and right lane line markings were detected for 86.1% of the route.

#### 4.3.2. CERPM

Figure 21a shows the CERPM detections for test route 3 with low curvature. Figure 21b shows the CERPM reference path along with the ground truth.

#### 4.3.3. LC Performance

The MSE for CERPM and Mobileye controller path to the ground truth is shown in Figure 22a and Figure 22b, respectively. The ground truth data were converted into local coordinates (NED) to match the local coordinates of the reference trajectory from Mobileye. MSE was computed in both X- and Y-directions. The MSE for CERPMs shown in Figure 22a in X-direction was 0.24 m and the MSE in Y-direction was 0.38 m. Figure 22b shows the MSE for the path taken using the Mobileye and the zones with successful detections have an MSE of 0.21 m in the X-direction and 0.45 m in the Y-direction.

## 5. Discussion

For test route 2, which has a steep curvature, it could detect the solid right lane marking for 81.7% and the dashed left lane marking for 6.7% of the route. For test route 3, which consists of low curvature, it was able to detect the solid right lane marking for 93% of the route and the dashed left lane marking for 86.1% of the route. Based on our observation, our new V2I CERPM technology yields better results when compared to Mobileye in both detection and trajectory generation for both test scenarios. Using the CERPM technology a trajectory could be generated successfully for both steep and low curvature test scenarios. Both technologies have their shortcomings, which are discussed in the next section. Table 1 shows the overall MSE magnitude in both the *x* and *y* directions for Mobileye and CERPM routes.

The Mobileye needs clear visible lane lines on both sides of the lane. During testing, it was noticed that the Mobileye struggled to identify the lane markings on either side of the lane on portions of the route with steep curvature, patchy lane markings with uneven colors, and shadows cast from the trees and surroundings that caused improper detections. Additionally, the Mobileye lacks information for lane markings beyond a distance of 31 m [32]. Whenever the curvature exceeded ±0.016 m^−1^, the Mobileye did not detect lane markings on either side of the lane. Our observation suggests that the Mobileye system performed better in terms of detections in road sections with low curvature as compared to steep curvature. A few drawbacks with our new CERPM technology were that, during the experiment, we observed that it was difficult to receive all CERPM signals simultaneously with a low-fidelity receiver and we are actively working on procuring and testing additional receiver options on the vehicle. Line-of-sight is crucial for a strong CERPM signal, similar to various wireless communication technologies. Although the CERPMs have some temporary shortcomings, they provide a preview of about 300 m that helps in predicting future trajectories when compared to Mobileye. Even in scenarios where there are signal dropouts or missing CERPMs, data processing techniques maintain detection performance. CEPRMs can be utilized in unmarked roads and varied lighting and weather conditions. Iterative improvements could lead to better detections, enabling further technological expansion. If CERPMs are to be scaled up in the future, this technology will essentially replace the current raised pavement markers. Each state’s Department of Transportation (DOT) already has maintenance plans in place for RPMs, so it will not require the development of new methods to install and maintain this technology. The installation of this technology will take time due to the number of CERPMs needed to cover the entire country, but depending on the use of this technology and its deployment, the process could be streamlined. To maintain the safety and security of this technology, we will explore the formal methods for implementing secure network protocols and information exchange, such as IEEE 802.11p and C-V2X [33]. The addition of data encryption and Rx to Tx authentication methods will help to ensure the safety of the data transmitted between the CERPM and the vehicle [34].

## 6. Conclusions

This study expands on our previously established work on CERPMs for real-world on-road V2I capabilities such as lane centering. The proposed work introduced a new method of vehicle control using perception information provided by our newly developed CERPMs. This new method of obtaining perception information was compared with a commercially available off-the-shelf computer vision solution, the Mobileye 630 Development Kit. It was observed that the Mobileye did not perform well on roads with sharp curves, varying lighting conditions, and inadequate lane markings. On the other hand, CERPMs can gather data up to a range of 350 m from the vehicle, enabling us to predict the vehicle’s future trajectory. In general, the Mobileye could not identify lane markings, and performance degraded when the road’s curvature exceeded ±0.016 m^−1^. Additionally, a controller was developed to perform LC using the CERPMs and Mobileye, respectively.

This study has successfully demonstrated that by using V2I information and leveraging our CERPM technology, we can obtain accurate and computationally inexpensive future lane information, which can then be used for AV perception and control. The CERPM information is not dependent on lane line visibility or lighting conditions as it does not depend on vision-based infrastructure. It can be successfully implemented for ADAS or AV features such as LC. This technology can effectively address the challenging drawbacks that come with commercial off-the-shelf vision providers used for drivable region detection required for ADAS or AVs. Iterative improvements are planned for the CEPRM hardware and software to enable large-scale testing. More advanced receiver technology and potentially using cellular integration could also be explored for future work. Fusing the information from the CERPMs with other on-vehicle sensors could improve the overall system performance for LC. Additionally, the CERPMs preview could be used to implement a more robust control algorithm, such as model predictive control. This study does not extensively delve into the economic feasibility of the CERPM technology, as it is not the primary focus. However, upon primary examination of the economic feasibility, it is important to note that the cost of setting up RSUs can amount to thousands of dollars at each intersection. HD maps necessitate a probe vehicle and a substantial amount of engineering hours. In contrast, regular raised pavement markers cost a couple of dollars and can be adapted using off-the-shelf IoT boards. When produced in large quantities for extensive road networks, the cost of CERPMs would be significantly lower.

## Figures and Tables

**Figure 1 sensors-24-02327-f001:**
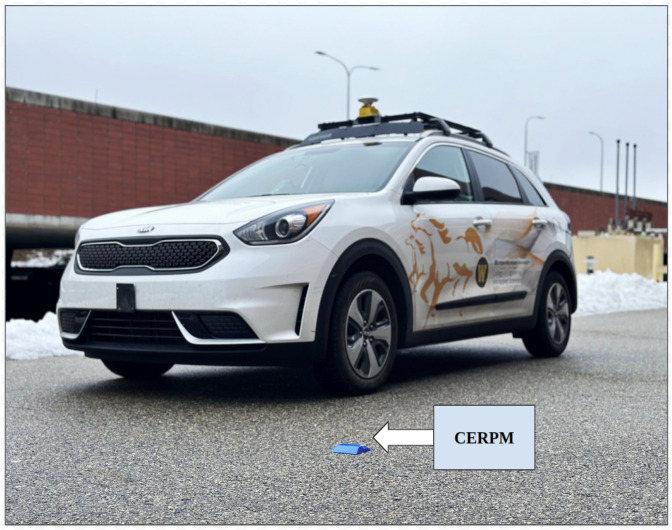
Energy-efficient autonomous vehicle (EEAV) research platform along with an RSU CERPM unit pictured at Western Michigan University.

**Figure 2 sensors-24-02327-f002:**
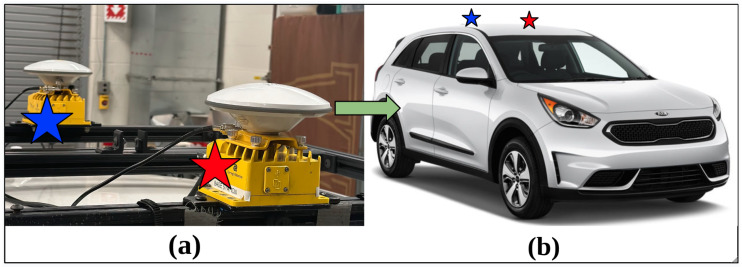
Dual SwiftNav GNSS setup on the EEAV research vehicle platform: (**a**) Placement of the SwiftNav Duro Inetrial RTK represented with the red star and placement of the SwiftNav Duro RTK represented with the blue star. The setup from (**a**) is mounted on the vehicle roof as shown with the green arrow. (**b**) Placement of the Dual GNSS RTK Setup on the EEAV research vehicle.

**Figure 3 sensors-24-02327-f003:**
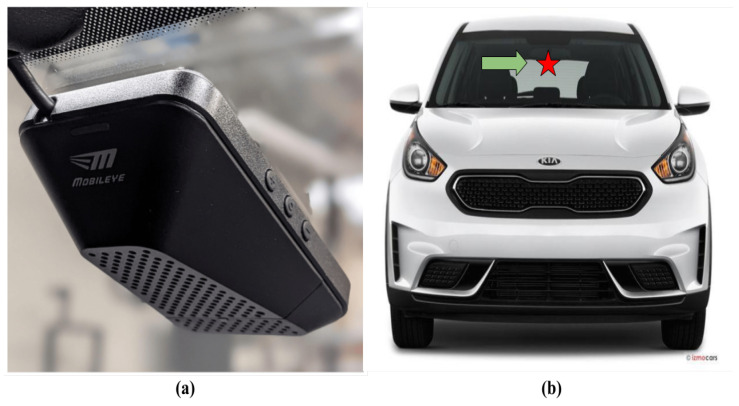
Mobileye 6 Development Kit mounting position on the EEAV research vehicle platform. (**a**) The Mobileye 6 Development Kit. (**b**) Mounting position of the Mobileye 6 Development Kit on the windshield. The green arrow shows the mounting position of the Mobileye on the vehicle windshield.

**Figure 4 sensors-24-02327-f004:**
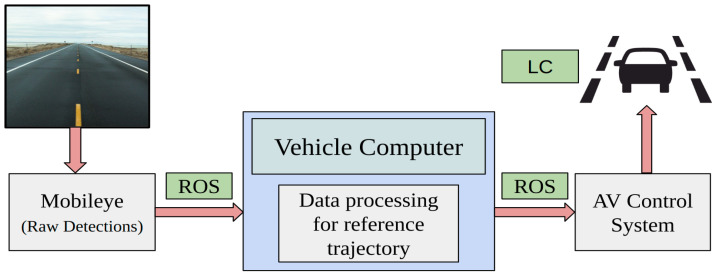
The Mobileye raw detections are sent from the sensor to the in-vehicle perception system using the ROS environment. The perception system carries out data processing for generating the reference trajectory, which is then made available for the controls subsystem.

**Figure 5 sensors-24-02327-f005:**
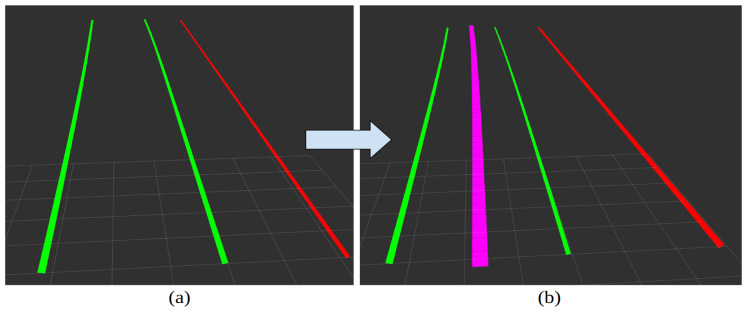
Visualization information obtained through the as_tx/lane_markers topic displayed in RViz for a single lane road: (**a**) Starting from the left, the green line represents the left lane line marking, the second green line is the right lane line marking, and the red line is the lane boundary. (**b**) The center of the current lane, the reference trajectory, is shown using the purple line. The processing goes from (**a**) to (**b**) shown using the blue arrow.

**Figure 6 sensors-24-02327-f006:**
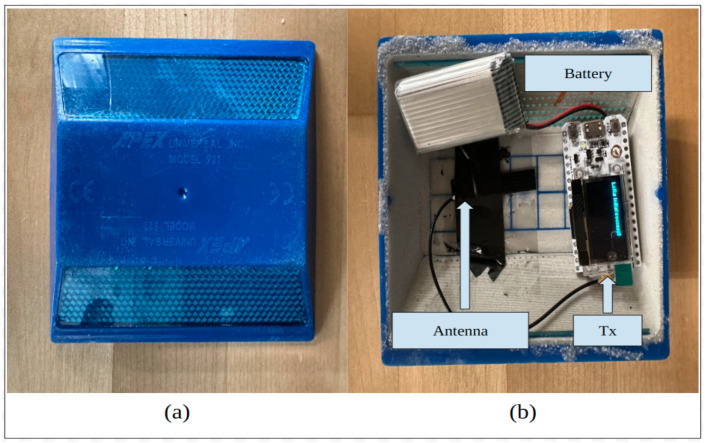
Modifying a regular RPM to serve as a CERPM, which will serve as the Tx and placed on the road for geospatial data transmission (**a**) standard raised pavement marker. (**b**) Modified raised pavement marker that includes the IoT development board, this would serve as the CERPM.

**Figure 7 sensors-24-02327-f007:**
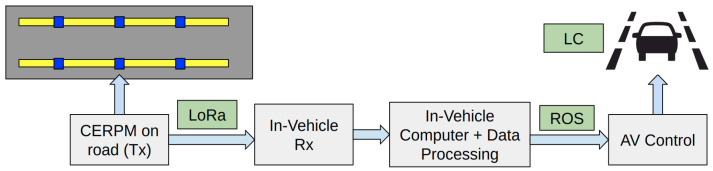
The CERPM Data Routing Plan: The on-road Tx’s information is received by the in-vehicle Rx and sent to the in-vehicle computer using the custom ROS driver. This information is further processed by the perception subsystem. The output from the perception system is used for control calculation and lastly used to apply vehicle control.

**Figure 8 sensors-24-02327-f008:**
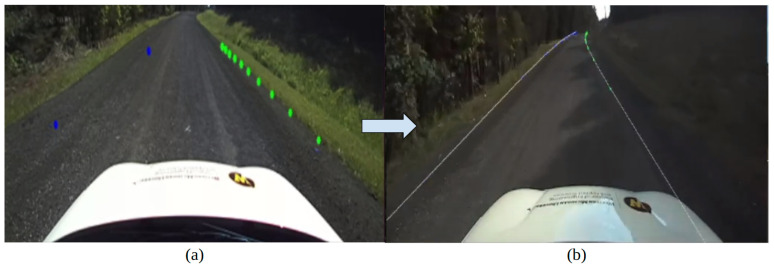
Projection of CERPMs on the camera feed for visual verification and pseudo drivable region generation. (**a**) Projection of four CERPMs on the camera feed. Two non-interpolated CERPMs were placed at a separation of 40 feet denoting the left lane boundary shown in blue markers. Two interpolated CERPMs on the right lane boundary shown with green markers. (**b**) Using 10 CERPMs on each lane boundary to generate pseudo lane lines for drivable region creation shown using the blue arrow from (**a**) to (**b**).

**Figure 9 sensors-24-02327-f009:**
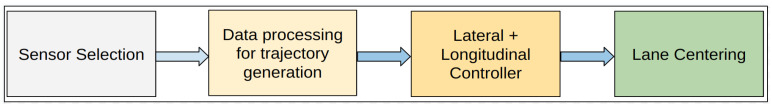
Flow diagram for using perception data to apply vehicle control. Either Mobileye or CERPMs are used to obtain the inputs for the perception subsystem. The processed data from the perception subsystem is then passed on to the control subsystem. The control output is used for LC.

**Figure 10 sensors-24-02327-f010:**
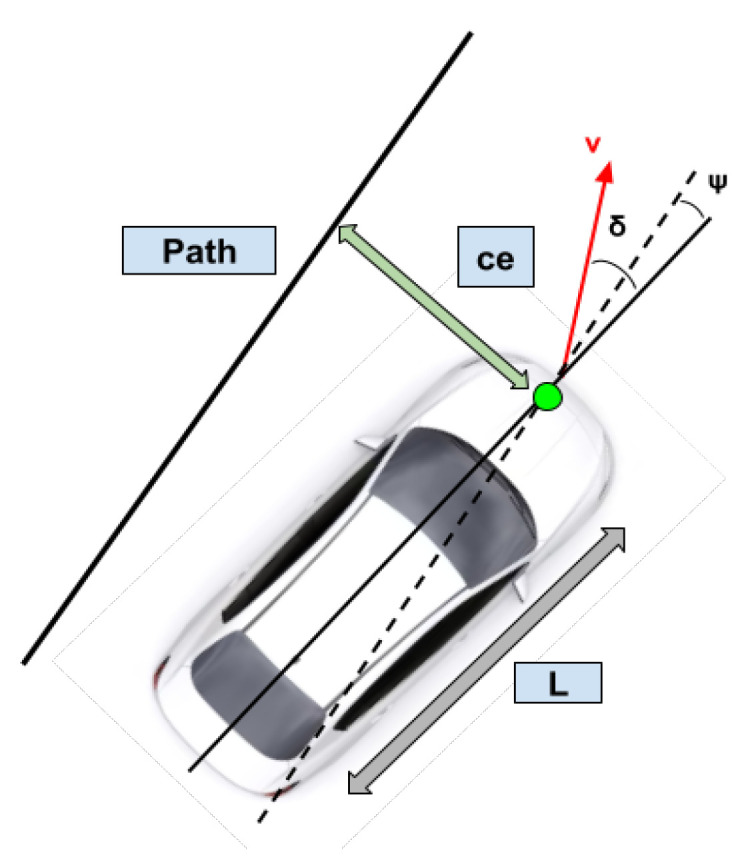
Given the cross-track error and heading error, the Stanley controller minimizes both errors and aligns the vehicle to the reference path. The dashed lines show the path parallel to the reference path, the solid black line that goes through the center of the vehicle shows the vehicle heading, the green marker shows the center of the front axle and the arrow shows the cross-track error.

**Figure 11 sensors-24-02327-f011:**
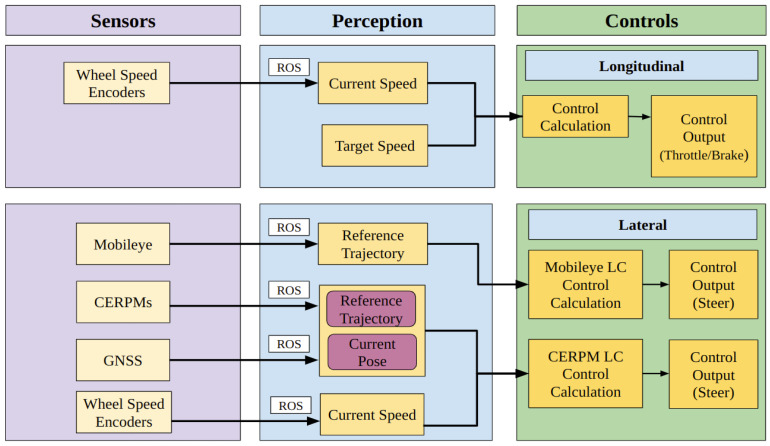
For longitudinal control, the wheel speed encoders provide information to the perception subsystem and is further used for painting the target speed. Similarly, for lateral control, the Mobileye and the CERPMs are used to perform LC using the same control subsystem.

**Figure 12 sensors-24-02327-f012:**
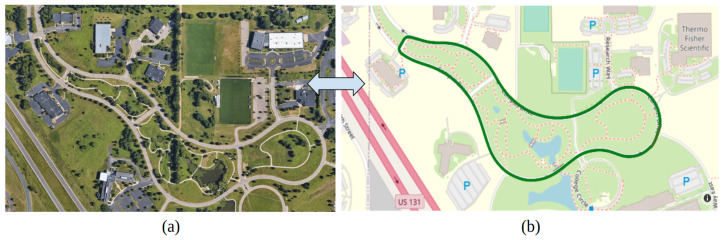
Test route at Western Michigan University. (**a**) Parkview Campus Drive Map. (**b**) Fixed route consisting of two-lane roads used for testing. Mapped for our previous study [28].

**Figure 13 sensors-24-02327-f013:**
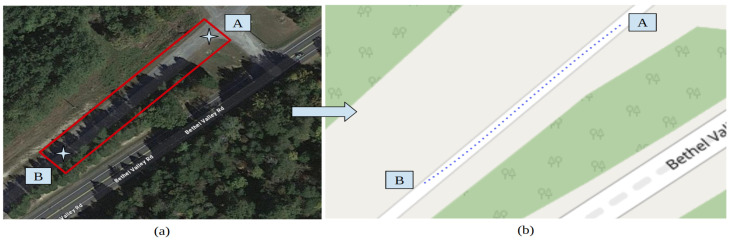
Test route 1 at Oak Ridge National Laboratory Main Campus. (**a**) Test route A to B at Old Bethel Valley Road–Oak Ridge. (**b**) Successful trajectory generation using 10 CERPMs.

**Figure 14 sensors-24-02327-f014:**
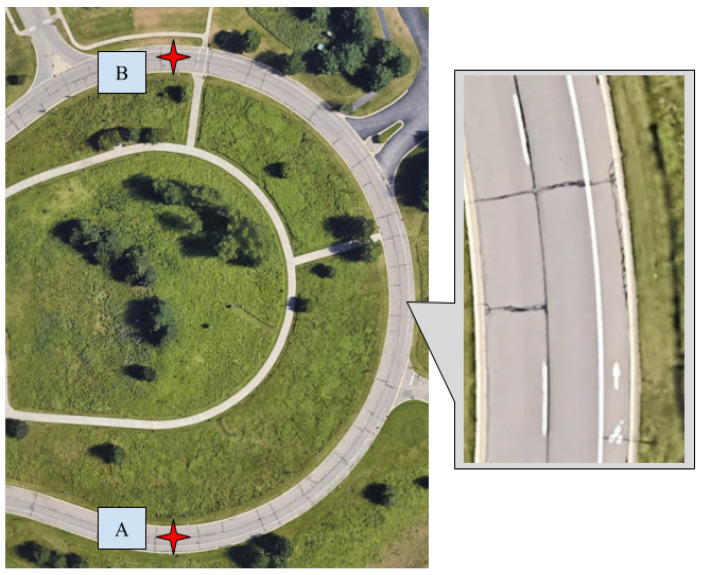
Test route 2 at Western Michigan University for steep curvature scenario. The outermost lane consists of lane markings on both sides of the lane, which would be utilized for the LC using Mobileye and CERPMs. Point A was the starting position for the test route and B was the end of the route.

**Figure 15 sensors-24-02327-f015:**
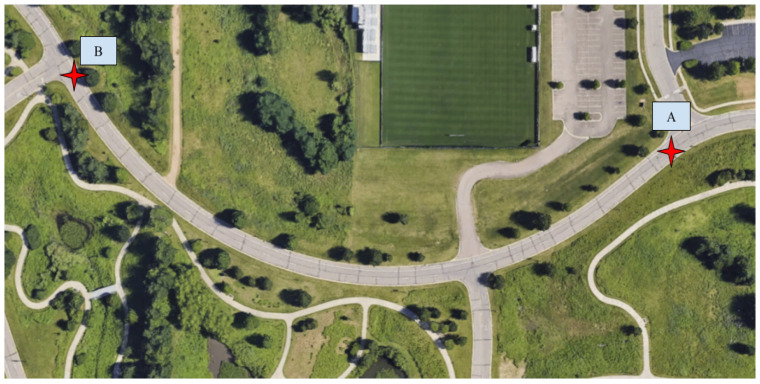
Test route 3 at Western Michigan University for low curvature scenario. The outermost lane consists of lane markings on both sides of the lane, which would be utilized for the LC using Mobileye and CERPMs. Test was conducted from point A to point B.

**Figure 16 sensors-24-02327-f016:**
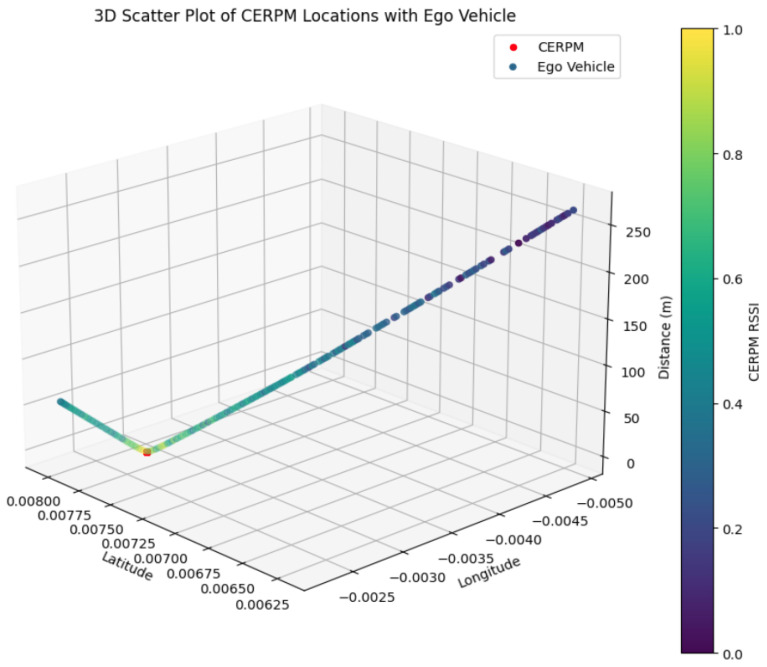
This plot shows the RSSI profile between the on-vehicle Rx and the on-road CERPM (Tx). The red data point is the position of the CERPM on the road, and the line on the Z-axis the RSSI value with different distances from the ego-vehicle. The colorbar maps the RSSI values.

**Figure 17 sensors-24-02327-f017:**
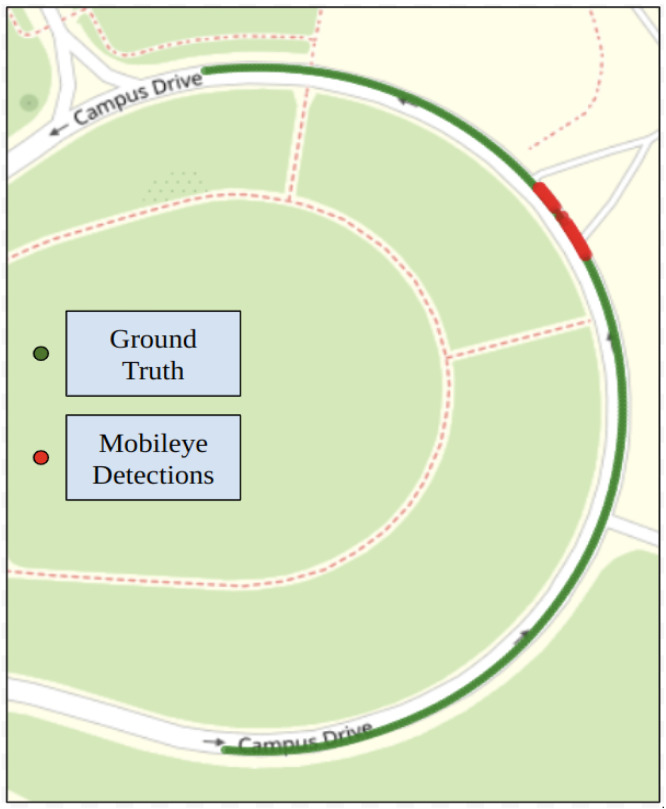
Mobileye detections and ground truth data for test route 2.

**Figure 18 sensors-24-02327-f018:**
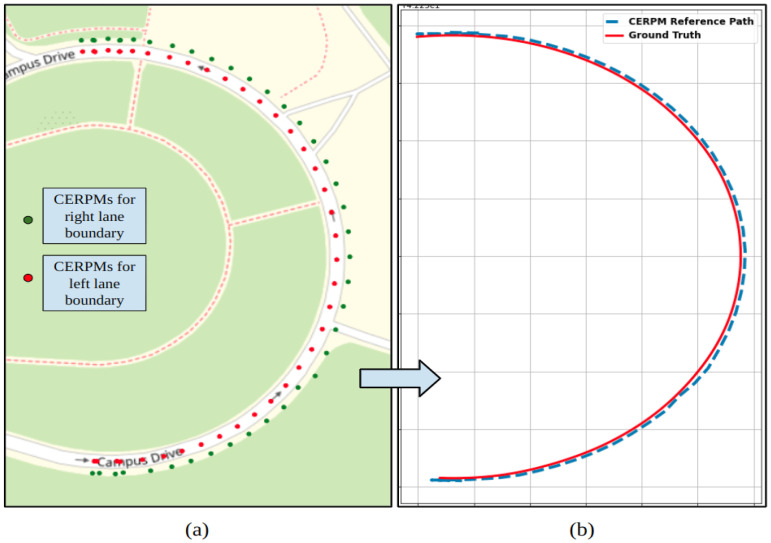
CERPM detections for test route 2 with steep curvature. (**a**) Raw detections from CERPMs. (**b**) Processed CERPM information, providing the reference trajectory for vehicle control along with ground truth data.

**Figure 19 sensors-24-02327-f019:**
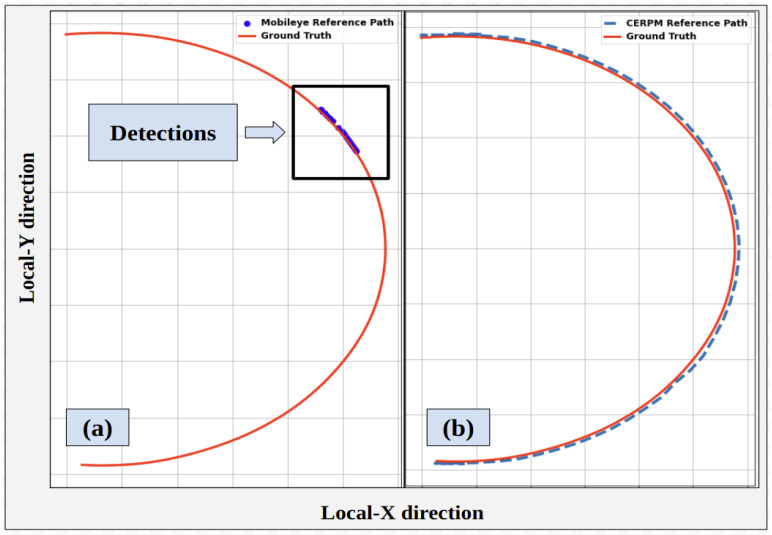
LC path using the two different systems for test route 2 compared to ground truth. (**a**) Controller path using Mobileye. (**b**) Controller path using CERPMs.

**Figure 20 sensors-24-02327-f020:**
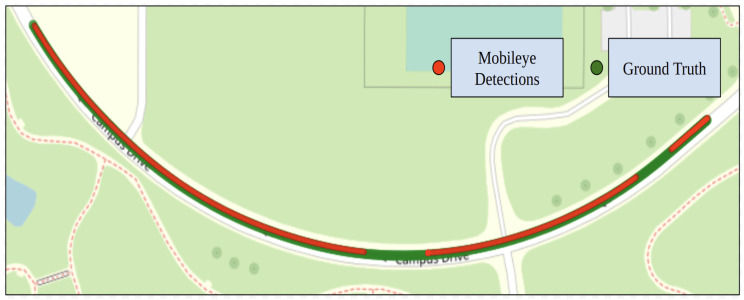
Mobileye detections and ground truth data for test route 3.

**Figure 21 sensors-24-02327-f021:**
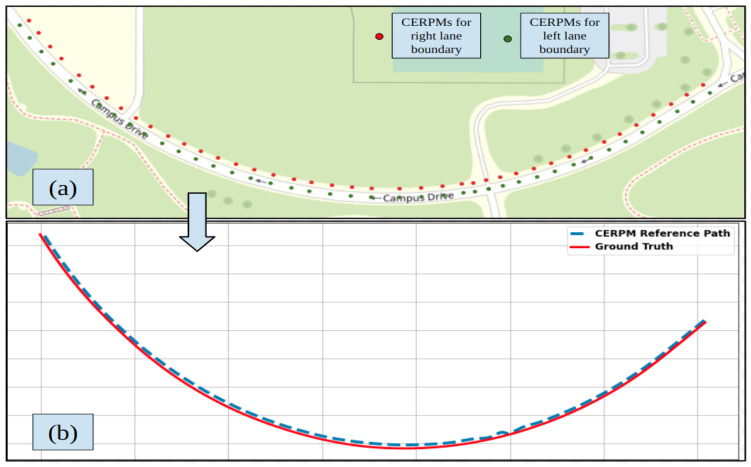
CERPM detections for test route 3 with low curvature. (**a**) Raw detections from CERPMs. (**b**) Processed CERPM information, providing the reference trajectory for vehicle control along with ground truth data.

**Figure 22 sensors-24-02327-f022:**
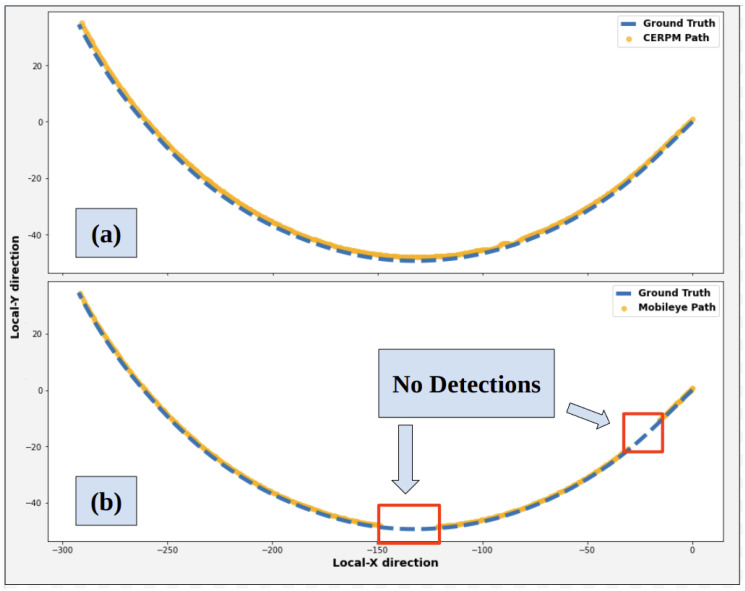
LC path taken by the vehicle using the two different systems for test route 3 compared to ground truth. (**a**) Controller path using CERPMs. (**b**) Controller path using Mobileye.

**Table 1 sensors-24-02327-t001:** Summary of CERPM and Mobileye performance using the Stanley controller for LC. The overall magnitude of the error from both the X- and Y-direction.

Type	CERPM	Mobileye
Test Route 2 MSE	0.42 m	N/A ^1^
Test Route 3 MSE	0.38 m	0.41 m

^1^ For test route 2 (steep curvature), the controller could not be enabled as there were no detections from the Mobileye to pass to the control subsystem.

## Data Availability

The data presented in this article are not readily available because the data is a part of an ongoing study. Requests to access the datasets should be directed to parth.kadav@wmich.edu.

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
