# Peer review of "Automated Lane Centering: An Off-the-Shelf Computer Vision Product vs. Infrastructure-Based Chip-Enabled Raised Pavement Markers"

_sensors, 2024, doi:10.3390/s24072327_

Round 1

Reviewer 1 Report

Comments and Suggestions for Authors

Contribution/Summary: The main contribution of the research is the proposal of a new Infrastructure Information Source (IIS) called Chip-Enabled Raised Pavement Markers (CERPMs), which provide environmental data to autonomous vehicles (AVs) while reducing the computational load and energy usage of the AVs. The CERPMs transmit geospatial information and speed limits directly to nearby vehicles using the Long Range Wide Area Network (LoRaWAN) protocol. This information is then used by the perception subsystem of the AV to generate a reference trajectory for vehicle control.

Comments/Suggestions:

  • Provide more details about the methodology used for evaluating the detection performance of both the CERPMs and the Mobileye system, such as the specific test routes and conditions considered.
  • Include a comparison of the cost-effectiveness of implementing CERPMs versus other infrastructure-based solutions, such as High Definition maps and Road Side Units.
  • Discuss the limitations and potential challenges of implementing CERPMs, such as the need for widespread adoption and maintenance of the markers along road lane lines.
  • Provide more information about the custom ROS driver created for publishing information from the CERPMs to the perception subsystem, including the specific functionalities and integration process.
  • Discuss the potential impact of CERPMs on the overall energy efficiency of AVs, including quantitative data on the reduction in vehicle energy use achieved by using CERPMs.
  • Include a comparison of the detection performance of CERPMs in different environmental conditions, such as bad weather, low light, and poor lane markings, to assess their reliability and performance in challenging scenarios.
  • Discuss the potential scalability and feasibility of implementing CERPMs on a larger scale, considering factors such as the required infrastructure and coordination with state Departments of Transportation.
  • Provide insights into the future research directions and potential advancements in CERPM technology, such as exploring additional data transmission capabilities or integrating CERPMs with other sensor systems for enhanced perception.
  • The authors may add some recent references about the use of formal methods and validation techniques for ensuring the security of Automative Systems.
  • For instance, they may include the following references and others:
    a. https://www.mdpi.com/2078-2489/14/12/666
    b. https://ieeexplore.ieee.org/document/8971034
Comments on the Quality of English Language

Can be improved.

Reviewer 2 Report

Comments and Suggestions for Authors

This paper proposes a new infrastructure information source that provides environmental data to autonomous vehicles while also reducing vehicle computing load and energy usage. The following modifications are recommended:

1. Has the communication delay issue involved in the use of new infrastructure information sources been considered? If you have considered it, please explain in detail.

2. According to lines 245-247, Figure 7 cannot clearly describe the difference with Mobileye, and it is recommended to redraw it.

3. It is recommended that the order of the coordinate axis directions in lines 192-193 be explained in the order of XYZ.

4. The 'RViz' in line 199 is not consistent with the 'RViz' explained in Figure 5. It is advisable to standardize the two for consistency.

5. In Figure 5, the lane line boundary is far away from the right lane line. Could this be attributed to the selection of a two-lane road for the test? It would be beneficial for the author to clarify this in the text.

6. Please label Figures 5, 8, 12, 13, 18, and 21 according to (a) and (b).

7. Regarding the electrical connection sequence on lines 223-224, please clearly indicate whether the battery powers the antenna and circuit board?

8. Please explain whether the brake is used in line 330.

9. Please provide a legend for Figure 18.

10. Table 1 is not mentioned in the text, so it is recommended to supplement it completely.

Comments on the Quality of English Language

The English sentences are fluent and clear, with only a few spelling problems. The sentences are professional and highly readable.

Round 2

Reviewer 1 Report

Comments and Suggestions for Authors

The authors considered my comments and suggestions. Good luck.

Comments on the Quality of English Language

A final proofread would be useful.